# Vaping and mental health: A cross-sectional study among university students in Bangladesh

Farah Sabrina[1,2*], Mohammad Delwer Hossain Hawlader[1,3], Md. Nur Alam[1], Md. Farhan Ibne Faruq[1], Farah Parisha Bhuiyan[4], Biswajit Banik[5], Muhammad Aziz Rahman[5,6]

1 Department of Public Health, North South University, Dhaka, Bangladesh, 2 Public Health Promotion and Development Society (PPDS), Dhaka, Bangladesh, 3 NSU Global Health Institute (NGHI), North South University, Dhaka, Bangladesh, 4 Institute of Medicine, The University of Chester, Chester, United Kingdom, 5 Institute of Health and Wellbeing, Federation University Australia, Berwick, Victoria, Australia, 6 Faculty of Public Health, Universitas Airlangga, Surabaya, Indonesia

* sabrinafarahbds40@gmail.com

## Abstract

### Introduction

Vaping has continued to rise, besides smoking, among youth in Bangladesh in recent years. While there has been evidence of the impact of vaping on physical health, studies focusing on mental health specifically from the South Asian context are almost non-existent. Therefore, this study examines the association between vaping and a spectrum of mental health issues, such as psychological distress, depression, and anxiety, among university students in Bangladesh.

### Methods

A cross-sectional study with undergraduate students, aged 18−25 years, from seven universities in Bangladesh, was conducted. Data were collected using a web-based questionnaire. Data on smoking, vaping, and dual use were collected. Psychological distress was measured by using the K10 scale, while the CES-D10 and GAD-7 scales assessed depression and anxiety, respectively. Univariate and multivariate logistic regression analyses determined the relevant associations. Adjusted Odds Ratios (AORs) and 95% Confidence Intervals (CIs) were reported.

### Results

Of the 1615 study participants, males and females were distributed equally, and 54% were from two private universities. Findings revealed that one in six (15.4%, n = 248) participants were currently vaping. Exclusive current smokers were 6.2%, exclusive vape users were 6.5%, and dual users were 8.9%. Among vape users, the prevalence of psychological distress (80.5% vs. 76.6%) and depression (63.8% vs. 60.8%) was higher among dual users compared to current vape users; anxiety was similar (56.9% vs. 57.6%) in both groups. After adjusting potential confounders,

**Data availability statement:** All relevant data within the manuscript. Data are available for the corresponding author upon reasonable request.

**Funding:** The author(s) received no specific funding for this work.

**Competing interests:** The authors have declared that no competing interests exist.

current vaping was associated with drinking alcohol (AOR 11.43, 95% CI 7.41–17.63) and used of recreational drugs (AOR 4.29, 95% CI 2.36–7.79) However, dual use was associated with higher depression (AOR 1.93; 95% CI 1.04–3.57) and without a preexisting mental health condition was associated with severe anxiety (AOR 2.00, 95% CI 1.25–3.20).

## Conclusion

The study underscores a concerning impact on mental health amongst the young group of the population who were vaping, specifically among the dual users. Besides raising awareness, university-based tobacco cessation support and counselling should be considered a student well-being support strategy.

## 1. Introduction

Vaping has gained remarkable popularity worldwide in recent years, especially among current smokers and young adults [1,2]. The World Health Organization (WHO) reported that the global prevalence of vaping was 1.9% for both sexes, with 2.3% prevalence among males and 1.5% prevalence among females in 2024 [3]. According to the 2018 National Youth Tobacco Survey (NYTS), 3.6 million adolescents in the USA were involved in vaping, with a considerable increase in use recorded between 2017 and 2018. The percentage of adolescents vaping in the USA climbed from 1.5% to 20.8% from 2011 to 2018 [4]. Similar upward trends have been observed in other high-income countries. For instance, in Canada, the proportion of young adults who vaped nearly tripled from 5.7% in 2014 to 15% in 2019 [5]. In Australia, current vaping almost tripled among the population, with a nearly four-fold increase among those aged 18–24 between 2019 (5.3%) and 2022−23 (20.6%) [6]. In Asian contexts, the prevalence of vaping among university students varies considerably, but shows an overall increasing trend, ranging from 4.6% in China [7] to 23% in India [8] and 34.5% in Malaysia [9]. However, research from South Asian settings remains limited, despite the growing popularity of vaping among youth, possibly due to the gradual adoption of Western lifestyle influences [10]. In Bangladesh, the Global Adult Tobacco Survey (GATS) 2017 reported a low vaping prevalence of 0.2% among both males and females [11]. More recent data, however, suggest a sharp rise. A study conducted during the COVID-19 pandemic found a vaping prevalence of 31.3% among young adults [12]. The considerable discrepancy between these findings may reflect differences in target populations, sampling technique, data collection methods, and the evolving social acceptability of vaping rather than an actual epidemic-level surge [12]. Previous studies also indicate that dual use of traditional cigarettes and vaping is common, with many users perceiving vaping as less harmful than smoking [13].

Vaping is increasingly recognized as a public health concern due to its association with multiple adverse health outcomes. It has been linked to nicotine addiction, respiratory damage (including conditions such as "bronchiolitis obliterans" or "popcorn lung"), and cardiovascular risks such as elevated heart rate, hypertension, and

atherosclerosis [14]. Moreover, most vapes contain nicotine, which can impair brain development and pose significant health risks to adolescents, young adults, pregnant women, and fetuses [15,16]. Beyond physical harm, emerging evidence suggests that vaping may also affect mental health. Vaping potentially disrupts brain development and fosters unhealthy dependence, especially in adolescents, due to nicotine and other harmful compounds [17]. Vaping can potentially worsen symptoms of stress and anxiety by creating a bidirectional relationship with substance use [18]. In addition, vaping behavior among adolescents and young adults has been associated with an increased risk of substance abuse, such as cannabis, marijuana, or cocaine [14].

A theoretical explanation for the relationship between vaping and mental health can be grounded in biological and psychological mechanisms. Previous molecular-level research showed that nicotine exposure from cigarettes affects neurotransmitter systems such as dopamine and serotonin, which play key roles in mood regulation by deregulating neurotransmitter pathways, causing anxiety and depression [19]. Chronic nicotine use dysregulates these pathways, increasing vulnerability to depression and anxiety. Vaping, containing nicotine, will have a similar mechanism of impact on mental health. Additionally, smoking reduces the activity of the antioxidant enzyme, paraoxonase, which is associated with major depressive disorder due to oxidative stress [20]. Empirical evidence also supports this link; a U.S. CDC study found that 42.1% of students who currently used vape reported moderate-to-severe symptoms of depression and anxiety, compared with 21.0% of students who did not vape (P < .001) [21]. These findings, together with the known neurobiological effects of nicotine, provide a theoretical basis for hypothesizing that vaping may contribute to mental health problems through neurochemical imbalance and oxidative stress pathways [21].

Studies exploring the mental health correlates of vaping remain limited in low and middle-income countries (LMICs). For instance, a recent study in Thailand found that current and dual vape users were more likely to report symptoms of depression and anxiety compared to non-vapers [22]. However, most prior research has focused primarily on the prevalence, motivations for use, and physical health consequences of vaping, with relatively little attention to its mental health implications [8]. To address this gap, the present study aimed to examine the association between vaping and mental health outcomes. Specifically, focusing on psychological distress, anxiety, and depression among undergraduate university students in Bangladesh. This study explored how behavioral factors (vaping), psychological factors (mental health symptoms), and social (youth lifestyle and peer influence) factors interacted within this emerging public health issue.

## 2. Materials and methods

### 2.1. Study design and settings

This was a cross-sectional study that continued from 1st February 2024–31st October 2024. It was conducted among undergraduate students from seven universities across Bangladesh. Among the seven universities, two were private, and five were public, and were conveniently selected based on geographic representation and accessibility through the investigators' collaboration networks. Both private universities and two public universities were located in the capital city of Dhaka, and the other three public universities were selected from the three large metropolitan areas of Chittagong, Sylhet, and Khulna in Bangladesh.

### 2.2. Study population

Students who were 18–25 years of age, pursuing Bachelor's degrees in the selected universities, and could respond to an English questionnaire online, were invited to participate in this study.

### 2.3. Sampling

The sample size was calculated using Open Epi [23]. There was an absence of prior nationally representative data on vaping among university students. A conservative estimate of 30% prevalence of vaping was adopted based on published

South Asian studies [8]. Considering that prevalence along with 95% confidence intervals (CIs), and 80% power, the estimated minimum sample size was 323. Convenient sampling was used to recruit the study participants.

## 2.4. Data collection

After obtaining the ethics approval, an online link to the web-based questionnaire was developed in Google Forms. The survey invitation with the online link and QR code was shared on social media platforms, emails, and text messages with the students in those selected universities. Teachers across different disciplines also circulated the survey link and QR code during the study period. Flyers with survey links and QR codes were also displayed in those universities' cafeterias, students' lounges, and libraries. Before accessing the questionnaire, participants were provided with study-related information outlining the purpose, procedures, and confidentiality measures. Only those who voluntarily provided informed consent were able to proceed with the survey. We conducted a pilot study with a small number of university students once the questionnaire was prepared to check the language of the survey. We made necessary modifications in languages before finalising the final version of the questionnaire. The survey was anonymous and self-administered without adding any personal data to minimize response bias. Participants were assured of their confidentiality. The Google Form was configured to allow only one submission per email to avoid double submission. The Google Form required completion of all mandatory fields, and participants could not proceed to the next section without completing each part. Therefore, there were no missing responses, minimizing the possibility of missing data. Before analysis, all responses were screened for completeness and duplicate entries.

## 2.5. Study tools

A structured survey questionnaire was used. The outcome variable of this study was a spectrum of mental health issues, such as psychological distress, depression, and anxiety, which were assessed using validated and reliable tools. Psychological distress was assessed by the K-10 scale (The Kessler Psychological Distress) [24]. It included ten questions about emotional states, each with a five-level response scale. The K-10 scale had each item graded on a scale of 1("none of the time") to 5 ("all of the time"), with scoring from a minimum score of 10 and a maximum score of 50. Higher scores indicated greater distress.

The Center for Epidemiologic Studies Depression Scale–10 (CESD-10) was used to measure depressive symptoms [25,26]. Each item was rated from 0 ("rarely or never") to 3 ("almost all of the time"), yielding a total score between 0 and 30; higher scores indicated more severe depressive symptoms.

The Generalized Anxiety Disorder Scale (GAD-7) was used to assess anxiety severity [25]. The GAD-7 had seven questions with a 4-point Likert scale. It included seven items rated from 0 ("not at all") to 3 ("nearly every day"), with total scores ranging from 0 to 21; higher scores indicated greater anxiety severity.

Other variables included in the questionnaire were smoking habits, vaping patterns, the reason for vape use, sociodemographic variables including age, gender, socioeconomic status, living status, student information like university type, academic year, and behavioral issues like drinking alcohol and use of recreational drugs.

## 2.6. Data analyses

Data were analyzed using STATA (version 17). Descriptive analyses were used to describe the study variables. Categorical variables were expressed as frequencies and percentages. Means and standard deviations were used for continuous variables like age and each scale (K-10, CESD-10, and GAD Scale). To conduct inferential analyses, the scoring of the K-10 tool was grouped into low (score 10–15) and moderate to very high (score 16–50) psychological distress, CESD-10 into no depression (score <10), and depression (score >10), and GAD-7 into minimal to mild anxiety (score 0–9) and moderate to severe anxiety (score 10–21). The association between vaping and all categorical independent variables was examined by using a chi-square test. Fisher's exact tests were used if a categorical variable had fewer than five

observations in a cell. To examine whether the observed associations were independent of underlying psychological vulnerabilities, we conducted a subgroup analysis restricted to students without pre-existing mental health conditions [27]. Bivariate logistic regression was used to identify the association between vaping and mental health outcomes. Then, multivariate logistic regression was used; the model included all independent variables as confounders, which were significant in the chi-square tests at $p < 0.20$ [28,29]. In the logistic regression analyses, multicollinearity of the independent variables was also verified by looking at the regression coefficients' standard errors (SEs). The results of bivariate and multivariate logistic regression were presented as odds ratios (ORs) and adjusted odds ratios (AORs), respectively, 95% confidence intervals (CIs) were also reported. All statistics were tested using a two-sided test, and a p-value of <0.05 was considered statistically significant.

### 2.7. Ethics

Ethical approval was obtained from the Institutional Review Board of North South University in Bangladesh (2020/OR-NSU/IRB/1104). All the data were stored in a password-protected computer of the first author and were only accessible to the research team. All the responses were encrypted during submission and storage. The survey invitation's plain-language statement mentioned the anonymity of participation. Participation was voluntary, and participants could withdraw at any time during data collection.

### 3. Results

(a) Descriptive Results:

A total of 1,615 undergraduate students participated in this study. The characteristics of the study population were detailed in Table 1. Among the study participants, about two-thirds (62.2%) were aged 18–22 years. More than half of the participants (54.0%) were from two private universities. One in nine participants (11.1%) self-reported their mental health status as fair to poor. More than two-thirds of participants (69.2%) experienced moderate to very high levels of psychological distress, more than half (53.0%) exhibited depression, and almost half (49.4%) were dealing with anxiety.

Table 2 shows, one in six participants (15.3%) reported being current vape users, and (6.2%) were smoking currently. One in eleven (9.0%) were dual-users (used cigarettes and vaping simultaneously). Half of the vape users (50.0%) used them occasionally and had nicotine in the e-juice (58.0%). One-third (37.50%) mentioned curiosity as the primary reason for vaping, and almost a quarter (25.0%) perceived vaping as less addictive than cigarettes.

Among current vape users, the prevalence of psychological distress (76.6%), depression (60.8%), and anxiety (57.6%) was notably high. Among dual users, psychological distress was 80.5%, depression was 63.8%, and anxiety was 56.9% (Table 3).

(b) Inferential Results:

Associations between current vaping and various socio-demographic factors through multivariate analyses were presented in Table 4. After controlling for all potential confounders(Gender, Academic year, Socioeconomic status, Drinking alcohol, Use of recreational drugs, and Diagnosed mental health problem) it showed that study participants were in the third to final year (AOR 1.66, 95% CI 1.06–2.60), who were from medium to high socioeconomic status (AOR 3.05, 95% CI 1.99–4.70), who used to drink alcohol daily (AOR 11.43, 95% CI 7.41–17.63,), and who used recreational drugs daily (AOR 4.29, 95% CI 2.36–7.79,), who diagnosed mental health problem(OR = 0.33; 95% CI 0.23–0.47; $p < 0.001$) had higher odds of using vaping.

Study participants with moderate to very high levels of psychological distress, depression, and moderate to severe anxiety had higher odds of using vapes in the univariate analyses. Vaping was associated with psychological distress (OR 1.55; 95% CI 1.13–2.13), depression (OR 1.45; 1.10–1.91), and anxiety (OR 1.48;1.12–1.94), although the significance did not remain after adjusting for potential confounders (Table 4).

**Table 1. Characteristics of the study population.**

| Characteristics | Total, n (%) |
| --- | --- |
| **Total study participants** | **1,615** |
| **Age** | |
| 18–22 years | 1,004 (62.2) |
| 23–30 years | 611 (37.8) |
| **Gender** | |
| Male | 802 (49.7) |
| Female | 813 (50.3) |
| **University type** | |
| Public university | 745 (46.1) |
| Private university | 870 (53.9) |
| **Academic year** | |
| First to second year | 920 (57.0) |
| Third to final year | 695 (43.0) |
| **Socioeconomic status** | |
| **Low income:** (household income BDT 10,000–30,000, equivalent to USD 81.61–244.84 per month) | 588 (36.4) |
| **Medium to high-income:** Medium (household income BDT 30,001–80,000, equivalent to USD 244.84–652.92), High (household income more than BDT 80,000, equivalent to USD 652.92) | 1,027 (63.6) |
| **Marital status** | |
| Unmarried | 1,506 (93.3) |
| Married | 109 (6.7) |
| **Living status** | |
| With family | 748 (46.3) |
| Living alone/＋hostel+ shared flat [not living with family or living independently] | 867 (53.7) |
| **Financial contribution to the family** | |
| No, I am fully dependent on my family | 1,312 (81.2) |
| Yes, a part of my earnings goes to my family | 303 (18.8) |
| **Perception of current social life** | |
| Least Satisfied | 579 (35.9) |
| Satisfied | 1,036 (64.1) |
| **Comorbidity** | |
| No | 1,369 (84.8) |
| Yes | 246 (15.2) |
| **Drinking alcohol** | |
| No | 1,432 (88.7) |
| Daily | 14 (0.9) |
| Occasionally | 169 (10.5) |
| **Use of recreational drugs** | |
| No | 1,514 (93.7) |
| Daily | 16 (1.0) |
| Occasionally | 85 (5.3) |
| **Diagnosed mental health problem** | |
| No | 1,435 (88.9) |
| Yes | 180 (11.1) |

*(Continued)*

**Table 1.** (Continued)

| Characteristics | Total, n (%) |
|---|---|
| **K-10 scale** | |
| Low (score 10–15) | 498 (30.8) |
| Moderate to Very High (score 16–50) | 1,117 (69.2) |
| **CESD-10 scale** | |
| No Depression (score 0–9) | 757 (46.9) |
| Depression (score 10–30) | 858 (53.1) |
| **GAD-7 scale** | |
| Minimal to mild anxiety (score 0–9) | 817 (50.6) |
| Moderate to severe anxiety (score 10–21) | 798 (49.4) |

Table 5 indicates, dual use of vaping and cigarettes was significantly higher among study participants studying at private universities (AOR 2.10, 95% CI 1.19–3.72), being in the third to final year (AOR 2.50, 95% CI 1.30–4.82, P = 0.006), being male (AOR 0.19, 95% CI 0.1–0.33), belonging to medium to high socioeconomic status (AOR 5.54, 95% CI 2.72–11.26), drinking alcohol (AOR 52.97, 95% CI 26.22–107.0), using recreational drugs (AOR 5.74, 95% CI 2.13–15.4), and having depression (AOR 1.93, 95% CI 1.04–3.57). Conversely, those who had been diagnosed with mental health issues (OR 0.27, 95% CI 0.17–0.42) were associated with less likely of dual use.

Table 6 shows the factors associated with vaping among the students who did not have pre-existing mental health issues. After adjusting for potential confounders, studying at private universities (AOR 1.53, 95% CI 1.00–2.33), belonging to medium to high socioeconomic status (AOR 3.27, 95% CI 2.01–5.34), drinking alcohol (AOR 11.42, 95% CI 7.10–18.38), using recreational drugs (AOR 5.87, 95% CI 2.81–12.28), and students with moderate to severe anxiety (AOR 2.00, 95% CI 1.25–3.20) were more likely to vape. This subgroup analysis was exploratory and aimed to see if the associations held in students without prior mental health diagnoses; therefore, these findings were interpreted with caution.

## 4. Discussion

This cross-sectional study provided an overview of vaping prevalence and the association between vaping and psychological distress, depression, and anxiety among undergraduate students of public and private universities in Bangladesh. Our findings provided critical insights into the various factors associated with that association. We also examined the factors associated with the dual use of vaping and cigarettes. This study was possibly the first study from Bangladesh to examine the association between vaping and mental health status among university students in Bangladesh.

In this study, the prevalence of vaping among undergraduate university students in Bangladesh was 15.4%, with 8.9% of students identified as dual use of vaping and cigarettes. Similarly, in India, 23% of the young population were current vape users [8]. In contrast, these figures were lower than those reported in Malaysia, where 74.9% of students were vapers, and 40.3% were dual users [9]. From 2014 to 2018, the prevalence of current vape users among U.S. adults aged 19–28 rose from 9% to 17% [30]. Another study among university students in the U.S. reported a vaping use rate of 12.8% [31]. Additionally, compared with studies in North America, among students of similar age, 25.2% were daily vape users [32]. In Canada, youth vape users nearly tripled between 2014 and 2019, increasing from 5.7% to 15% [5]. Variations across countries may be explained by differences in regulation, accessibility, affordability, and cultural norms regarding smoking environments [18,33].

In the current study, the main reasons for vaping were curiosity (37.5%), peer pressure (26.2%), and those who had greater concern with social trends (18.5%) were more likely to vape. This finding was consistent with findings from a previous study in Bangladesh [12,13]. In addition, in Malaysia, research showed that, among university students who

**Table 2. Smoking patterns of the study participants.**

| Characteristics | Total, n (%) |
|---|---|
| **Total Participants** | **1,615** |
| **Smoking Status** | |
| Never smoking (cigarettes or vaping) | 1,233 (76.4) |
| Smoking only (Current smokers, never vaped) | 100 (6.2) |
| Current vape users with no prior smoking | 22 (1.4) |
| Current vape users with previous smoking history | 82 (5.1) |
| Dual use (both smoking and vaping) | 144 (8.9) |
| Ex-smokers, never vaped | 34 (2.1) |
| **Smoking pattern** | |
| No smoking | 1,225 (75.8) |
| Daily smoking | 182 (11.3) |
| Occasional smoking | 127 (7.9) |
| Intention to quit smoking | 47 (2.9) |
| Quit smoking <1year ago | 18 (1.1) |
| Quit smoking >1year ago | 16 (0.9) |
| **Vaping** | |
| No | 1,367 (84.6) |
| Yes (vaping only+dual use) | 248 (15.4) |
| **Reasons for vaping** | |
| Curiosity | 93 (37.5) |
| Inspired by friends | 65(26.2) |
| Current trends | 46(18.5) |
| Curiosity/Inspired by friends/ Current trends/Others | 44 (17.7) |
| **Types of vaping liquid** | |
| With nicotine | 144 (58.1) |
| Without nicotine (only flavors) | 85 (34.3) |
| Without nicotine (no flavors) | 19 (7.9) |
| **Nicotine concentration (mg)** | |
| Unknown | 10 (6.9) |
| ≤6mg nicotine | 35 (24.3) |
| ≤9mg nicotine | 44 (30.5) |
| ≤12mg nicotine | 23 (15.9) |
| ≤16mg nicotine | 32 (22.2) |
| **Perceived that vaping was less addictive than cigarettes** | |
| No | 213 (13.2) |
| Yes | 402 (24.9) |
| Don't know/ Not sure | 1,000 (61.9) |
| **Family members have a habit of smoking** | |
| No | 1,080 (66.8) |
| Father | 379 (23.9) |
| Mother | 1 (0.1) |
| Sister | 152 (9.4) |
| Brother | 3 (0.2) |
| **Perception of effective ways to quit smoking** | |
| Counselling | 305 (18.9) |
| Vaping | 56 (3.5) |

*(Continued)*

**Table 2.** (Continued)

| Characteristics | Total, n (%) |
|---|---|
| Nicotine replacement drug | 29 (1.8) |
| Nicotine Patch/ Nicotine Lozenge | 23 (1.4) |
| Counselling + Nicotine patch + Traditional medicine | 120 (7.4) |
| Vaping + Nicotine lozenge + Nicotine replacement drug+ Nicotine patch | 74 (4.6) |
| Others | 214 (13.3) |
| Not Sure/ Don't know | 794 (49.2) |
| **Physical effects of using a vape** | |
| Dizziness | 27 (10.9) |
| Cough | 51 (20.6) |
| Headache | 20 (8.1) |
| Chest pain/ Shortness of breath | 15 (6.1) |
| Insomnia | 14 (5.6) |
| Decrease Appetite/ Weight loss | 49 (19.8) |
| Cough + Dizziness + Shortness of breath | 48 (19.4) |
| No/ Don't know | 24 (9.7) |

**Table 3. Prevalence of psychological distress, depression, and anxiety among vape users and dual users.**

| | Current vape users, n (%) | Dual users, n (%) |
|---|---|---|
| **K-10 scale** | | |
| Low (score 10–15) | 58 (23.3) | 28(19.4) |
| Moderate to very high (score 16–50) | 190 (76.6) | 116 (80.5) |
| **CESD-10 scale** | | |
| No Depression (score 0–9) | 97 (39.1) | 52(36.1) |
| Depression (score 10–30) | 151 (60.8) | 92 (63.8) |
| **GAD-7 Scale** | | |
| Minimal to mild anxiety (score 0–9) | 105(42.3) | 62(43.6) |
| Moderate to severe anxiety (score 10–21) | 143(57.6) | 82(56.9) |

were exclusive vape users, the main reasons given for vape use were current trend (28.8%), mood disorder (27.8%), and social influence (25.1%) [9]. Disposable vapes gained market share recently due to the affordability and curiosity of users, specifically among young people, high nicotine content, and a variety of flavors, marketing strategies of tobacco companies, and Misconceptions about vaping being a safer alternative to cigarettes also played a role. These findings were interpreted as associations with the motivation of vaping [7,34].

Male students were significantly more likely to vape than female students. A Chinese study showed that males were more likely to vape (6.6% of males and 2.7% of females) [35]. Similarly, A study from Malaysia revealed that 42.3% of vape users were male. In contrast, 3.9% of females used vapes [9]. This gender gap likely reflected sociocultural norms in Asia, within which tobacco use among women was considered socially unacceptable [36].

Furthermore, students from medium to high socioeconomic backgrounds were more likely to have vaped than those from lower-income families. A Malaysian study showed that higher household income was associated with a greater likelihood of vaping among university students. Similarly, people with higher household incomes in China were 1.54 times

**Table 4. Factors associated with vaping among undergraduate university students in Bangladesh.**

| Characteristics | Current vaping | | Unadjusted analysis | | | Adjusted analysis* | | |
|---|---|---|---|---|---|---|---|---|
| | Yes | No | | | | | | |
| | n (%) | n (%) | p | ORs | 95% CIs | p | AORs | 95% CIs |
| **Age** | | | | | | | | |
| 18–22 years | 111 (11.1) | 893 (88.9) | <0.001 | Ref. | | 0.606 | Ref. | |
| 23–30 years | 137 (22.4) | 474 (77.6) | | 2.32 | 1.76 - 3.05 | | 1.12 | 0.72 - 1.75 |
| **Gender** | | | | | | | | |
| Male | 186 (23.2) | 616 (76.8) | <0.001 | Ref. | | <0.001 | Ref. | |
| Female | 62 (7.6) | 751 (92.4) | | 0.27 | 0.20 - 0.37 | | 0.28 | 0.19 - 0.42 |
| **Academic year** | | | | | | | | |
| First to second year | 103 (11.2) | 817 (88.8) | <0.001 | Ref. | | 0.027 | Ref. | |
| Third to final year | 145 (20.9) | 550 (79.1) | | 2.09 | 1.58 - 2.75 | | 1.66 | 1.06 - 2.60 |
| **Socioeconomic status** | | | | | | | | |
| Low income | 46 (7.8) | 542 (92.2) | <0.001 | Ref. | | <0.001 | Ref. | |
| Medium to high-income | 202 (19.7) | 825 (80.3) | | 2.88 | 2.05 - 4.04 | | 3.05 | 1.99 - 4.70 |
| **Marital status** | | | | | | | | |
| Unmarried | 230 (15.3) | 1276 (84.7) | 0.720 | Ref. | | Not selected in the multivariate model | | |
| Married | 18 (16.5) | 91 (83.5) | | 1.09 | 0.64 - 1.85 | | | |
| **Living status** | | | | | | | | |
| With Family | 131 (17.5) | 617 (82.5) | 0.020 | Ref. | | 0.209 | Ref. | |
| Living alone+ hostel+ shared flat | 117 (13.5) | 750 (86.5) | | 0.73 | 0.56 - 0.96 | | 0.79 | 0.55 - 1.13 |
| **Financial contribution to the family** | | | | | | | | |
| No, I am fully dependent on my family | 177 (13.5) | 1135 (86.5) | <0.001 | Ref. | | 0.810 | Ref. | |
| Yes, a part of my earnings goes to my family | 71 (23.4) | 232 (76.6) | | 1.96 | 1.44 - 2.67 | | 0.94 | 0.61 - 1.46 |
| **Perception of current social life** | | | | | | | | |
| Least Satisfied | 87 (15.0) | 492 (85.0) | 0.783 | Ref. | | Not selected in the multivariate model | | |
| Satisfied | 161 (15.5) | 875 (84.5) | | 1.04 | 0.78 - 1.38 | | | |
| **Comorbidities** | | | | | | | | |
| No | 202 (14.8) | 1167 (85.2) | 0.115 | Ref. | | Not selected in the multivariate model | | |
| Yes | 46 (18.7) | 200 (81.3) | | 0.75 | 0.52 - 1.07 | | | |
| **Drinking alcohol** | | | | | | | | |
| No alcohol | 121 (8.4) | 1311 (91.6) | <0.001 | Ref. | | <0.001 | Ref. | |
| Yes (Daily + Occasionally) | 127 (69.4) | 56 (30.6) | | 24.57 | 17.05 −35.41 | | 11.43 | 7.41 - 17.63 |
| **Use of recreational drugs** | | | | | | | | |
| No | 174 (11.5) | 1340 (88.5) | <0.001 | Ref. | | <0.001 | Ref. | |
| Yes (Daily + Occasionally) | 74 (73.3) | 27 (26.7) | | 21.10 | 13.21 −33.70 | | 4.29 | 2.36 - 7.79 |
| **Diagnosed mental health problem** | | | | | | | | |
| No | 191 (13.3) | 1244 (86.7) | <0.001 | Ref. | | 0.021 | Ref. | |
| Yes | 57 (31.7) | 123 (68.3) | | 0.33 | 0.23 - 0.47 | | 0.55 | 0.33 - 0.91 |
| **K-10 scale** | | | | | | | | |
| Low (score 10–15) | 58 (11.6) | 440 (88.4) | 0.006 | Ref. | | 0.555 | Ref. | |
| Moderate to very high (score 16–50) | 190 (17.0) | 927 (83.0) | | 1.55 | 1.13 - 2.13 | | 1.15 | 0.72 - 1.83 |
| **CESD-10 scale** | | | | | | | | |
| No Depression (score 0–9) | 97 (12.8) | 660 (87.2) | 0.008 | Ref. | | 0.259 | Ref. | |
| Depression (score 10–30) | 151 (17.6) | 707 (82.4) | | 1.45 | 1.10 - 1.91 | | 1.26 | 0.83 - 1.91 |
| **GAD-7 Scale** | | | | | | | | |

*(Continued)*

**Table 4.** (Continued)

| Characteristics | Current vaping | | Unadjusted analysis | | | Adjusted analysis* | | |
|---|---|---|---|---|---|---|---|---|
| | **Yes** | **No** | | | | | | |
| | **n (%)** | **n (%)** | **p** | **ORs** | **95% CIs** | **p** | **AORs** | **95% CIs** |
| Minimal to mild anxiety (score 0–9) | 105 (12.9) | 712 (87.1) | *0.005* | Ref. | | *0.079* | Ref. | |
| Moderate to severe anxiety (score 10–21) | 143 (17.9) | 655 (82.1) | | *1.48* | *1.12 - 1.94* | | *1.46* | *0.95 - 2.24* |

** Stepwise multivariate logistic regression analysis was conducted. Variables with $p < 0.20$ in univariate analysis and those identified as potential confounders (gender, academic year, socioeconomic status, drinking alcohol, recreational drug use, and diagnosed mental health problem) were included in the initial model. Only variables that remained significant at $p < 0.05$ were presented in the final adjusted model. Adjusted odds ratios (AORs) and 95% confidence intervals (CIs) were shown.

more likely to have been vape users [9,37]. This might have been linked to greater purchasing power and easier access to vaping products through online platforms and retail outlets among individuals with higher disposable income [33]. Hence, individuals with higher socioeconomic status tended to exhibit a greater likelihood of vaping.

Dual use of cigarettes and vaping was strongly associated with an increased likelihood of experiencing depression, as highlighted by multiple studies conducted in different regions. In this study, multivariate logistic regression revealed that dual users were 1.93 times more likely to be depressed than non-vapers, which might have significant implications for public health. In addition, another study in Thailand also revealed that dual users were 2.30 times more likely to experience moderate to severe depression symptoms compared to non-vape users [22]. Another study in the USA found that both vape and traditional cigarette users were associated with higher depressive symptoms than single-product use, except when disposable vape and cigarettes were both used infrequently [38]. These results suggested that dual use might have reflected greater nicotine exposure, stronger dependence, and existing emotional distress, which might have explained the consistent association with depression observed across different regions.

The possible links between vaping and mental health issues might have been due to the biological effects of nicotine on mood and neurotransmitter systems. Nicotine stimulates the release of dopamine, serotonin, and norepinephrine neurotransmitters that regulate mood, attention, and arousal [22,38]. Young individuals might have been more vulnerable to nicotine's impact, as exposure during brain development could have increased susceptibility to mood and anxiety problems. Some vapes delivered nicotine more effectively or in higher concentrations, which could have exacerbated these effects. Additionally, the varying chemical composition of e-liquids and additives might have contributed to different mental health outcomes [14]. Repeatedly using a vape to cope with stress or anxiety could have led to an unhealthy dependence on the device, exacerbating existing mental health issues or even contributing to the development of new ones. In turn, vape use could have worsened their mental health [14,15]. These findings warranted strict public health intervention in regulating the use of various chemical compounds in e-cigarettes.

Alcohol and recreational drug use were high among the participants who reported vaping and dual smoking in this study. The odds of drinking alcohol were 11 times higher among non-vapers compared to vapers. In addition, in a large cross-sectional study involving young adults in the USA, one of the reputed universities found a significant link between alcohol use and vaping. They found that Alcohol usage, particularly binge drinking, has the strongest association with vape use. The odds of drinking alcohol were 1.36 times higher for non-vapers [31]. Furthermore, this study found that participants were 4.29 times more likely to use recreational drugs. These findings align with trends observed for other substances, such as alcohol and cannabis, closely mirroring the results of this study [39]. Some other studies in the USA have found that alcohol use, particularly binge drinking, was strongly linked to vape use [40,41]. These findings suggested that vaping might have occurred within a broader spectrum of risk-taking behaviors and substance use patterns among young adults. It was also possible that alcohol and drug use acted as confounding factors in the observed association between

Table 5. Factors associated with dual use (smoking and vaping) among undergraduate university students in Bangladesh.

| Characteristics | Dual use (smoking and vaping) | | Unadjusted analysis | | | Adjusted analysis* | | |
|---|---|---|---|---|---|---|---|---|
| | No | Yes | | | | | | |
| | n (%) | n (%) | p | ORs | 95% CIs | p | AORs | 95% CIs |
| **Age** | | | | | | | | |
| 18–22 years | 828 (92.3) | 69 (7.7) | <0.001 | Ref. | | 0.703 | Ref. | |
| 23–30 years | 405 (84.4) | 75 (15.6) | | 2.22 | 1.56-3.14 | | 0.88 | 0.46-1.67 |
| **Gender** | | | | | | | | |
| Male | 504 (82.5) | 107 (17.5) | <0.001 | Ref. | | <0.001 | Ref. | |
| Female | 729 (95.2) | 37 (4.8) | | 0.23 | 0.16-0.35 | | 0.19 | 0.1-0.33 |
| **University type** | | | | | | | | |
| Public university | 593 (91.8) | 53 (8.2) | 0.011 | Ref. | | 0.010 | Ref. | |
| Private university | 640 (87.6) | 91 (12.4) | | 1.59 | 1.11-2.27 | | 2.10 | 1.19-3.72 |
| **Academic year** | | | | | | | | |
| First to second year | 760 (92.5) | 62 (7.5) | <0.001 | Ref. | | 0.006 | Ref. | |
| Third to final year | 473 (85.2) | 82 (14.8) | | 2.12 | 1.49-3.01 | | 2.50 | 1.30 - 4.82 |
| **Socioeconomic status** | | | | | | | | |
| Low income | 499 (95.2) | 25 (4.8) | <0.001 | Ref. | | <0.001 | Ref. | |
| Medium to high-income | 734 (86.0) | 119 (14.0) | | 3.23 | 2.07-5.05 | | 5.54 | 2.72 - 11.26 |
| **Marital status** | | | | | | | | |
| Unmarried | 1,151 (89.4) | 136 (10.6) | 0.615 | Ref. | | Not selected in the multivariate model | | |
| Married | 82 (91.1) | 8 (8.9) | | 0.82 | 0.39-1.74 | | | |
| **Living status** | | | | | | | | |
| With Family | 557 (88.1) | 75 (11.9) | 0.116 | Ref. | | Not selected in the multivariate model | | |
| Living alone+ hostel+ shared flat | 676 (90.7) | 69 (9.3) | | 0.75 | 0.53-1.07 | | | |
| **Financial contribution to the family** | | | | | | | | |
| No, I am fully dependent on my family. | 1,039 (91.6) | 95 (8.4) | <0.001 | Ref. | | 0.223 | Ref. | |
| Yes, a part of my earnings goes to my family. | 194 (79.8) | 49 (20.2) | | 2.76 | 1.89 - 4.02 | | 1.45 | 0.79 - 2.64 |
| **Perception of current social life** | | | | | | | | |
| Least Satisfied | 423 (88.7) | 54 (11.3) | 0.446 | Ref. | | Not selected in the multivariate model | | |
| Satisfied | 810 (90.0) | 90 (10.0) | | 0.93 | 0.78-1.11 | | | |
| **Comorbidities** | | | | | | | | |
| No | 1,054 (89.3) | 126 (10.7) | 0.513 | Ref. | | Not selected in the multivariate model | | |
| Yes | 179 (90.9) | 18 (9.1) | | 1.18 | 0.70-1.99 | | | |
| **Drinking alcohol** | | | | | | | | |
| No alcohol | 1211 (95.2) | 61 (4.8) | <0.001 | Ref. | | <0.001 | Ref. | |
| Daily alcohol (Occasionally+daily) | 22 (21.0) | 83 (79.0) | | 74.89 | 43.83-128 | | 52.97 | 26.22–107.0 |
| **Use of recreational drugs** | | | | | | | | |
| No drug | 1,222 (92.7) | 96 (7.3) | <0.001 | Ref. | | <0.001 | Ref. | |
| Daily drug (Occasionally+daily) | 11 (18.6) | 48 (81.4) | | 55.54 | 27.93-110.44 | | 5.74 | 2.13 - 15.48 |
| **Diagnosed mental health problem** | | | | | | | | |
| No | 1,136 (91.2) | 110 (8.8) | <0.001 | Ref. | | 0.024 | Ref. | |
| Yes | 97 (74.0) | 34 (26.0) | | 0.27 | 0.17-0.42 | | 0.42 | 0.20 - 0.89 |
| **K-10 scale** | | | | | | | | |
| Low (score 10–15) | 408 (93.6) | 28 (6.4) | <0.001 | Ref. | | 0.121 | Ref. | |
| Moderate to very high (score 16–50) | 825 (87.7) | 116 (12.3) | | 2.04 | 1.33-3.14 | | 1.72 | 0.86 - 3.41 |

*(Continued)*

**Table 5.** (Continued)

| Characteristics | Dual use (smoking and vaping) | | Unadjusted analysis | | | Adjusted analysis* | | |
|---|---|---|---|---|---|---|---|---|
| | No | Yes | | | | | | |
| | n (%) | n (%) | p | ORs | 95% CIs | p | AORs | 95% CIs |
| **CESD-10 scale** | | | | | | | | |
| No depression (score 0–9) | 603 (92.1) | 52 (7.9) | *0.004* | Ref. | | *0.037* | Ref. | |
| Depressed (score 10–30) | 630 (87.3) | 92 (12.7) | | *1.69* | *1.18-2.42* | | *1.93* | *1.04-3.57* |
| **GAD-7 scale** | | | | | | | | |
| Minimal to mild anxiety (score 0–9) | 648 (91.3) | 62 (8.7) | *0.031* | Ref. | | 0.414 | Ref. | |
| Moderate to severe anxiety (score 10–21) | 585 (87.7) | 82 (12.3) | | *1.46* | *1.03-2.07* | | 1.29 | 0.69-2.39 |

**Stepwise multivariate logistic regression analysis was conducted. Variables with $p < 0.20$ in univariate analysis and theoretically relevant confounders (gender, university type, academic year, socioeconomic status, alcohol consumption, recreational drug use, diagnosed mental health problem, and CESD-10 scale) were entered into the initial model. Only variables that remained significant at $p < 0.05$ were presented in the final adjusted model. Adjusted odds ratios (AORs) and 95% confidence intervals (CIs) were shown.

vaping and mental health, as both were independently related to psychological distress. Future longitudinal research was warranted to better understand the interrelationship between these behaviors and to clarify causal pathways.

After excluding participants with pre-existing mental health conditions, in the multivariable analysis, students with moderate to severe anxiety were found to be twice as likely to vape compared to those without anxiety symptoms. This suggests that even in the absence of diagnosed mental health disorders, vaping behavior may contribute to anxiety. Similarly, findings from other studies have indicated that individuals with higher odds of anxiety were 2.7 times more likely to engage in vaping [18,22]. People may have used vaping as a coping mechanism to manage stress or regulate emotions, aligning with previous evidence that linked to this study. Causal relationships could not be inferred, and the findings should have been interpreted with caution. Future longitudinal studies were warranted to better understand the temporal and causal pathways between anxiety and vaping behavior.

Despite its strengths in the large sample size of seven universities, this study had several limitations. At first, a convenient sampling was used that did not represent the entire university population in Bangladesh. Secondly, the cross-sectional nature of the study design meant the regression results could only be interpreted as providing insights into associations between vaping and a range of variables; the direction of the relationship between vaping and mental health conditions cannot be established. While psychological distress, depression, and anxiety were analyzed as predictors in this study, it was also possible that vaping behavior might have influenced mental health outcomes. Future longitudinal research will be warranted to clarify these temporal and causal relationships. Thirdly, there was a possibility of reporting bias since no objective measures were used to verify smoking or vaping status. Participants might have underreported their behaviors due to social desirability or stigma associated with vaping and mental health issues, even though the survey was anonymous. There was also a possibility that recall bias might have influenced participants' ability to accurately report their past behaviors or experiences. Additionally, the use of a convenience sample of university students might have introduced selection bias, limiting the generalizability of the results to the broader young adult population in Bangladesh. Despite those limitations, this was probably the 1st study in Bangladesh where the impact of vaping on mental health status was examined among young people. Future research with a nationally representative sample and a longitudinal study will add further value to inform tobacco control strategies and mental health support for young people in Bangladesh.

## 5. Conclusion

This study provides new evidence on the association between vaping and mental health status among young undergraduate university students in Bangladesh. The dual use of vaping and cigarettes was more concerning in this study. Factors

**Table 6. Factors associated with vaping among undergraduate university students in Bangladesh who did not have pre-existing mental health issues.**

| Characteristics | Use of vaping | | Unadjusted analysis | | | Adjusted analysis* | | |
|---|---|---|---|---|---|---|---|---|
| | No | Yes | | | | | | |
| | n (%) | n (%) | p | ORs | 95% CIs | p | AORs | 95% CIs |
| **Age** | | | | | | | | |
| 18–22 years | 809 (90.1) | 89 (9.9) | <0.001 | Ref. | | 0.896 | Ref. | |
| 23–30 years | 435 (81.0) | 102 (18.9) | | 2.13 | 1.56–2.89 | | 1.03 | 0.63-1.69 |
| **Gender** | | | | | | | | |
| Male | 572 (79.2) | 150 (20.8) | <0.001 | Ref. | | <0.001 | Ref. | |
| Female | 672 (94.3) | 41 (5.8) | | 0.23 | 0.16–3.33 | | 0.22 | 0.14- 0.34 |
| **University type** | | | | | | | | |
| Public university | 592 (89.8) | 67 (10.2) | <0.001 | Ref. | | 0.040 | Ref. | |
| Private university | 652 (84.0) | 124 (15.9) | | 1.68 | 1.22-2.30 | | 1.53 | 1.00-2.33 |
| **Academic year** | | | | | | | | |
| First to second year | 743 (90.1) | 82 (9.9) | <0.001 | Ref. | | 0.181 | Ref. | |
| Third to final year | 501 (82.1) | 109 (17.8) | | 1.97 | 1.44-2.68 | | 1.40 | 0.85–2.29 |
| **Socioeconomic status** | | | | | | | | |
| Low income | 491 (93.7) | 33 (6.3) | <0.001 | Ref. | | <0.001 | Ref. | |
| Medium to high-income | 753 (82.6) | 158 (17.3) | | 3.12 | 2.10-4.62 | | 3.27 | 2.01 −5.34 |
| **Marital status** | | | | | | | | |
| Unmarried | 1162 (86.9) | 175 (13.1) | 0.364 | Ref. | | Not selected in the multivari- | | |
| Married | 82 (83.7) | 16 (16.3) | | 1.29 | 0.74-2.26 | ate model | | |
| **Living status** | | | | | | | | |
| With Family | 560 (83.9) | 107 (16.0) | 0.005 | Ref. | | 0.070 | Ref. | |
| Living alone+ Hostel+ shared flat | 684 (89.1) | 84 (10.9) | | 0.64 | 0.47-0.87 | | 0.69 | 0.46-1.03 |
| **Financial contribution to the family** | | | | | | | | |
| No, I am fully dependent on my family. | 1036 (88.1) | 140 (11.9) | <0.001 | Ref. | | 0.958 | Ref. | |
| Yes, a part of my earnings goes to my family. | 208 (80.3) | 51 (19.7) | | 1.81 | 1.27–2.58 | | 1.01 | 0.62 −1.64 |
| **Perception of current social life** | | | | | | | | |
| Least Satisfied | 418 (86.7) | 64 (13.3) | 0.980 | Ref. | | Not selected in the multivari- | | |
| Satisfied | 826 (86.7) | 127 (13.3) | | 1.00 | 0.72-1.38 | ate model | | |
| **Comorbidities** | | | | | | | | |
| No | 1085 (86.9) | 163 (13.1) | 0.473 | Ref. | | Not selected in the multivari- | | |
| Yes | 159 (85.0) | 28 (14.9) | | 1.17 | 0.75-1.80 | ate model | | |
| **Drinking alcohol** | | | | | | | | |
| No alcohol | 1197 (92.3) | 100 (7.7) | <0.001 | Ref. | | <0.001 | Ref. | |
| Daily alcohol (Occasionally+daily) | 47 (34.1) | 91 (65.9) | | 23.17 | 15.42-34.81 | | 11.42 | 7.10-18.38 |
| **Use of recreational drugs** | | | | | | | | |
| No drug | 1227(89.4) | 146 (10.6) | <0.001 | Ref. | | <0.001 | Ref. | |
| Daily drug (Occasionally+daily) | 17 (27.4) | 45 (72.6) | | 22.24 | 12.40-39.88 | | 5.87 | 2.81 −12.28 |
| **K-10 scale** | | | | | | | | |
| Low (score 10–15) | 428 (88.9) | 53 (11.0) | 0.070 | Ref. | | 0.856 | Ref. | |
| Moderate to very high (score 16–50) | 816 (85.5) | 138 (14.5) | | 1.36 | 0.97-1.91 | | 1.04 | 0.63 −1.72 |
| **CESD-10 scale** | | | | | | | | |
| No depression (score 0–9) | 635 (88.2) | 85 (11.8) | 0.090 | Ref. | | 0.363 | Ref. | |
| Depressed (score 10–30) | 609 (85.2) | 106 (14.8) | | 1.30 | 0.95-1.76 | | 1.22 | 0.79-1.90 |

*(Continued)*

**Table 6.** (Continued)

| Characteristics | Use of vaping | | Unadjusted analysis | | | Adjusted analysis* | | |
|---|---|---|---|---|---|---|---|---|
| | No | Yes | | | | | | |
| | n (%) | n (%) | p | ORs | 95% CIs | p | AORs | 95% CIs |
| **GAD-7 scale** | | | | | | | | |
| Minimal to mild anxiety (score 0–9) | 687 (88.9) | 85 (11.0) | *0.006* | Ref. | | *0.004* | Ref. | |
| Moderate to severe anxiety (score 10–21) | 557 (84.0) | 106 (15.9) | | *1.53* | *1.13-2.08* | | *2.00* | *1.25-3.20* |

**Stepwise multivariate logistic regression analysis was conducted among participants without pre-existing mental health issues. Variables with $p < 0.20$ in univariate analysis and known confounders (gender, university type, socioeconomic status, alcohol consumption, recreational drug use, and GAD-7 scale) were included in the model. Variables that remained significant at $p < 0.05$ were retained in the final model. Adjusted odds ratios (AORs) and 95% confidence intervals (CIs) were shown.

associated with such associations should be considered for developing smoking cessation interventions as well as mental health support in university settings. In particular, mobile health (mHealth) approaches, such as app-based counselling, text message reminders, and digital behavioral support, could be leveraged to reach students conveniently and confidentially. Additionally, universities could strengthen on-campus counselling programs and peer-support initiatives to address stress, anxiety, and substance use. These results can guide university policies and enhance mental health interventions, fostering support systems such as confidential counselling services for students grappling with recreational drug use and psychological issues.

## Supporting information

**S1 File. Dataset_Vaping_1615_Sample_.**
(XLSX)

## Acknowledgments

We also acknowledge the valuable contributions of two student representatives, Bayan Shafique and Tahsin Firdaus for their dedicated support during the data collection process.

## Author contributions

**Conceptualization:** Farah Sabrina.

**Data curation:** Farah Sabrina, Md. Nur Alam, Md. Farhan Ibne Faruq, Farah Parisha Bhuiyan.

**Formal analysis:** Farah Sabrina, Muhammad Aziz Rahman.

**Methodology:** Farah Sabrina, Muhammad Aziz Rahman.

**Project administration:** Farah Sabrina.

**Resources:** Farah Sabrina, Md. Nur Alam, Md. Farhan Ibne Faruq, Farah Parisha Bhuiyan.

**Software:** Farah Sabrina.

**Supervision:** Farah Sabrina, Mohammad Delwer Hossain Hawlader, Muhammad Aziz Rahman.

**Validation:** Farah Sabrina, Mohammad Delwer Hossain Hawlader, Biswajit Banik, Muhammad Aziz Rahman.

**Visualization:** Farah Sabrina.

**Writing – original draft:** Farah Sabrina.

**Writing – review & editing:** Farah Sabrina, Mohammad Delwer Hossain Hawlader, Biswajit Banik, Muhammad Aziz Rahman.

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
