## [Editor Report · Decision Letter 0]

24 Jun 2025

Dear Dr. Sabrina,

Thank you for submitting your manuscript to PLOS ONE. After careful consideration, we feel that it has merit but does not fully meet PLOS ONE’s publication criteria as it currently stands. Therefore, we invite you to submit a revised version of the manuscript that addresses the points raised during the review process.

We look forward to receiving your revised manuscript.

Kind regards,

Ali Awadallah Saeed

Academic Editor

PLOS ONE

Additional Editor Comments:

Thank you for this important and timely study. I have two points of clarification that I believe would enhance the transparency and rigor of your methodology, before i send the manuscript to reviewers:

University Selection Criteria: Could you please elaborate on the criteria used to select the seven universities across Bangladesh? Specifically, it would be helpful to understand the rationale behind including two private and five public institutions.

Use of Standardized Assessment Tools: Kindly clarify whether permission or ethical approval was obtained for the use of the standardized tools in your study—namely the Kessler Psychological Distress Scale (K-10), the CESD-10, and the GAD-7. Indicating this would strengthen the ethical integrity of the research.
---

## [Author Response · Author response to Decision Letter 1]

6 Jul 2025

PONE-D-25-24214

Vaping and mental health: A cross-sectional study among university students in Bangladesh

PLOS ONE

Dear Dr. Sabrina,

Thank you for submitting your manuscript to PLOS ONE. After careful consideration, we feel that it has merit but does not fully meet PLOS ONE’s publication criteria as it currently stands. Therefore, we invite you to submit a revised version of the manuscript that addresses the points raised during the review process.

We look forward to receiving your revised manuscript.

Kind regards,

Ali Awadallah Saeed

Academic Editor

PLOS ONE

Response: Thank you for your comments. We addressed this in the revised manuscript. We followed the formatting sample of the author's affiliation and changed it. PLEASE SEE page 1, lines 18 to 19. (Revised Manuscript _20250706_clean file)

Response: Thank you for your comments. We addressed this in the revised manuscript. All of our ethics statements are only in the METHODS section, PLEASE SEE page 6, lines -173-176. (Revised Manuscript _20250706_clean file)

Additional Editor Comments:

Thank you for this important and timely study. I have two points of clarification that I believe would enhance the transparency and rigor of your methodology, before I send the manuscript to reviewers:

University Selection Criteria: Could you please elaborate on the criteria used to select the seven universities across Bangladesh? Specifically, it would be helpful to understand the rationale behind including two private and five public institutions.

Response: Thank you for your comments. We addressed this in the revised manuscript. we selected these seven universities across Bangladesh. We employed convenience sampling based on the known networks of the investigators to recruit seven universities. PLEASE SEE page no-4, line no-108, 109. (Revised Manuscript _20250706_clean file)

Use of Standardized Assessment Tools: Kindly clarify whether permission or ethical approval was obtained for the use of the standardized tools in your study, namely the Kessler Psychological Distress Scale (K-10), the CESD-10, and the GAD-7. Indicating this would strengthen the ethical integrity of the research.

Response: Thank you for your comments. We addressed this in the revised manuscript. These three scales, namely, the Kessler Psychological Distress Scale (K-10), the Center for Epidemiologic Studies Depression Scale (short form) (CESD-10), and the Generalized Anxiety Disorder scale (GAD-7), are freely available in public domains and do not require any permission for usage for scientific purposes. As such, no additional permission or alike was required for the usage of these scales in this study. Please refer to the following sources for confirmation.

1.https://www.hcp.med.harvard.edu/ncs/k6_scales.php

2.https://eprovide.mapi-trust.org/instruments/center-for-epidemiologic-studies-depression-scale-10

3.https://www.pfizer.com/news/press-release/press-release-detail/pfizer_to_offer_free_public_access_to_mental_health_assessment_tools_to_improve_diagnosis_and_patient_care

Furthermore, we obtained ethics approval from the Institutional Review Board (IRB) of North South University in Bangladesh, with the reference number 2020/OR-NSU/IRB/1104. This approval would have permitted the use of these standardized tools in this study. PLEASE SEE, page no -5,6. line no -148 to 153. (Revised Manuscript _20250706_clean file)
---

## [Decision Letter · Decision Letter 1]

29 Sep 2025

Dear Dr. Sabrina,

Thank you for submitting your manuscript to PLOS ONE. After careful consideration, we feel that it has merit but does not fully meet PLOS ONE’s publication criteria as it currently stands. Therefore, we invite you to submit a revised version of the manuscript that addresses the points raised during the review process.

We look forward to receiving your revised manuscript.

Kind regards,

Ali Awadallah Saeed

Academic Editor

PLOS ONE

Journal Requirements:

Reviewers' comments:

Reviewer's Responses to Questions

**Comments to the Author**

Reviewer #1: (No Response)

Reviewer #2: (No Response)

2. Is the manuscript technically sound, and do the data support the conclusions?

Reviewer #1: Partly

Reviewer #2: Partly

3. Has the statistical analysis been performed appropriately and rigorously?

Reviewer #1: No

Reviewer #2: Yes

4. Have the authors made all data underlying the findings in their manuscript fully available?

Reviewer #1: No

Reviewer #2: Yes

5. Is the manuscript presented in an intelligible fashion and written in standard English?

Reviewer #1: No

Reviewer #2: Yes

Reviewer #1: Comments to the Editor

Dear Editor,

Thanks for the opportunity to review this manuscript. Please find my comments below.

Introduction

The introduction contains several grammatical errors, awkward phrasings, and wordy constructions that make reading difficult. There are issues with sentence structure and clarity in multiple instances.

There is a significant inconsistency in reported vaping prevalence in Bangladesh (from 0.2% to 31.27%) without sufficient explanation of methodological differences between studies. This weakens credibility and may confuse readers.

Several ideas are repeated unnecessarily, such as the harmful effects of nicotine and health risks for pregnant women. These could be condensed to maintain reader engagement and clarity.

The paragraph progression lacks smooth transitions between global trends, regional context, and local findings. The narrative jumps between prevalence, physical effects, and mental health without clear thematic segmentation.

There is no theoretical basis or explanatory model provided to support the hypothesized relationship between vaping and mental health. This weakens the foundation for the research objectives.

While the introduction mentions a lack of studies from developing countries, it does not clearly articulate what specific aspect is missing in current literature or how this study uniquely addresses it.

Some claims, such as describing post-COVID vaping as an “epidemic,” are overstated or emotive without sufficient supporting evidence, affecting the academic tone of the introduction.

The objective of the study is presented at the end but could be more concisely stated and clearly linked to the identified research gap.

Method

Lack of information about how participants were distributed across universities and demographic strata raises concerns about selection bias.

No mention of any blinding, data validation, or procedures to reduce response bias in self-administered surveys.

The assumed 30% vaping prevalence is based on anecdotal evidence, which weakens the scientific rigor of the sample size justification.

Repetition occurs across several parts of the section (e.g., ethical approval mentioned twice, phrasing redundancies in informed consent).

Descriptions of K-10, CESD-10, and GAD-7 are overly long and not consistently formatted.

There's a factual inconsistency in referring to the CESD-10 but citing a source intended for the CESD-20 or CESD-R.

The rationale behind the logistic regression modeling approach and the threshold (p < 0.20) for variable inclusion lacks clarity and justification.

No detail is provided on handling missing data or response validation for online survey submissions.

Ethical approval and consent information are repeated unnecessarily.

Specific details about participant protections (e.g., data encryption or access controls) are not described.

There are formatting inconsistencies.

Results

Several statements repeat information already shown in tables (e.g., prevalence rates of psychological distress, depression, and anxiety among vape and dual users).

Repetition of percentages and exact numbers without analytical interpretation weakens narrative flow.

Terms such as “vaping only-1” and “vaping only-2” are unclear and inconsistently labeled across tables and text.

Typographical issues such as misplaced line breaks and disjointed headings (e.g., Table 2 and vaping descriptions) interrupt readability.

Data analysis leans mostly on descriptive statistics without a clear transition to inferential results.

The rationale for selecting variables into multivariate models is not consistently explained.

Tables show adjusted and unadjusted odds ratios, but variable inclusion criteria in multivariate models are only briefly referenced at the bottom of each table and inconsistently reported.

Some CIs and p-values do not align well with conclusions drawn (e.g., borderline significance interpreted as definitive associations).

There is a lack of clarity on whether mental health outcomes are predictors or outcomes of vaping, especially in subgroup analyses.

The role of diagnosed mental health conditions is inconsistently interpreted—sometimes associated with decreased vaping, other times not explained.

Subgroup analysis for individuals without pre-existing mental health conditions (Table 6) lacks a clear rationale and may inflate Type I error without proper correction or framing.

Transitions between narrative results and table interpretations are abrupt and sometimes fragmented.

Discussion

Key ideas such as the prevalence of vaping in Bangladesh and its rise during COVID-19 are repeated with slight rewording.

The association between vaping and mental health is revisited multiple times across different paragraphs without offering new insights.

Citations are presented back-to-back with little synthesis or interpretation.

External studies are described at length, but their connection to the study’s specific findings is not always clearly established.

Some statements imply causal links (e.g., “compulsion to follow trends” or “mental health could increase vaping”) despite the cross-sectional design.

Phrases like “which is a big public health concern” are emotionally charged and could be more neutrally framed.

Findings from univariate analysis are emphasized even when multivariate analysis showed no significance.

Contradictions occur when interpreting mental health variables as significant in some places but not in others without resolving the inconsistency.

Long paragraphs with multiple ideas (e.g., discussing sex, socioeconomic status, and regional comparisons) affect flow and readability.

Subtopics such as “dual use,” “alcohol and drug use,” and “socioeconomic status” are not clearly delineated.

Terms like “non-users” are inconsistently defined, sometimes meaning non-vapers and other times meaning non-smokers/vapers.

Occasional grammar and syntax issues (e.g., “Contiguity of smoking environments” or “the odds of drinking alcohol were 11 times higher for non-users”).

Explanations for non-significant results in multivariable analyses are brief and speculative.

The role of confounding variables is mentioned but not explored in depth (e.g., impact of alcohol/drug use on mental health and vaping).

Neurobiological explanations for the nicotine-mental health link are mentioned but lack detail or citation depth.

Limitations

Statements like “this warrants strict public health intervention” are not linked to specific policy or intervention models.

Biases like recall bias, selection bias, and the impact of social desirability are acknowledged but not critically examined.

No mention of potential underrepresentation of female users due to cultural factors.

Conclusion

The conclusion largely reiterates prior points without introducing a distinct final perspective.

Recommendations for university mental health policies are broad and not grounded in study findings or feasible actions.

Recommendation

Overall, the study has some merits but is not suitable for publication in its current form. I therefore recommend revision.

Reviewer #2: Dear editor, thank you for the kind invitation to review this paper.

Dear authors thank you for this great work. I ask the authors to address these comments or provide clarifications.

- First, there’s several studies that examine the use of vape and mental health even in developing countries. I advise the authors to focus their aim in literature (novel aspect) on Bangladesh settings, if there’s another novel aspect please correct me and consider it.

- Can you justify the selection of these 7 universities?

- In the study population you mentioned “who could respond to an English questionnaire” can you justify this? How is this not biased to the study population? As I know, the English language is not the official Bangladesh language.

- Can you add more context about the Open Epi and cite it.

- Your sample size was 323 and your final sample was about 1,600 in a data collection period of 10 months. Can you explain this? What was your aim longitudinal study?

**Do you want your identity to be public for this peer review?** For information about this choice, including consent withdrawal, please see our Privacy Policy

Reviewer #1: **Yes:** Dr Ngozika Esther Ezinne

Reviewer #2: **Yes:** Dr. Ahmad Mohammad Al Zamel

---

## [Author Response · Author response to Decision Letter 2]

21 Dec 2025

Reviewer #1: Yes: Dr Ngozika Esther Ezinne

Reviewer #2: Yes: Dr. Ahmad Mohammad Al Zamel

---

## [Editor Report · Decision Letter 2]

2 Jan 2026

Dear Dr.  Sabrina,

Thank you for submitting your manuscript to PLOS ONE. After careful consideration, we feel that it has merit but does not fully meet PLOS ONE’s publication criteria as it currently stands. Therefore, we invite you to submit a revised version of the manuscript that addresses the points raised during the review process.

We look forward to receiving your revised manuscript.

Kind regards,

Ali Awadallah Saeed

Academic Editor

PLOS One

Journal Requirements:

Additional Editor Comments:

Question to the authors:

The sample-size calculation assumes a 30% prevalence of vaping among university students in Bangladesh, described as anecdotal in the absence of available data. Could you clarify the scientific basis for selecting this value? Please indicate whether it was derived from published studies, pilot data, or a conservative estimation approach, and consider providing an appropriate reference or justification.

You may also use PLOS’s free figure tool, NAAS, to help you prepare publication quality figures: https://journals.plos.org/plosone/s/figures#loc-tools-for-figure-preparation

---

## [Author Response · Author response to Decision Letter 3]

20 Jan 2026

Response to Editor Comments:

Thank you for the insightful remarks. We appreciate the advice and comments, which undoubtedly improve the quality of our manuscript.

Editor Comment 1

Comment:

Response:

Thank you for your comment. We carefully reviewed all reviewer-recommended references. Relevant and scientifically appropriate publications have been incorporated into the manuscript. References that were not directly aligned with the study objectives or context were not included.

Editor Comment 2

Comment: Please review your reference list to ensure that it is complete and correct. If you have cited papers that have been retracted, please include the rationale for doing so in the manuscript text, or remove these references and replace them with relevant current references. Any changes to the reference list should be mentioned in the rebuttal letter that accompanies your revised manuscript. If you need to cite a retracted article, indicate the article’s retracted status in the References list and also include a citation and full reference for the retraction notice.

Response:

We carefully reviewed all the references for completeness, accuracy, compliance with journal guidelines, and retraction status using publisher websites, CrossRef, and PubMed records.

No retracted articles were identified among the cited references; therefore, no retraction notices were required to be added, and no references were removed for this reason.

Several technical corrections were made to improve the accuracy and integrity of the reference list, including the removal of duplicate entries. Specifically, References 34, 39, and 41 were identified as duplicates and have therefore been removed from the track-changed version. The references in the Discussion section have been reorganized to maintain correct order and consistency with the updated reference list, both the revised manuscript and the track-changed version.

There are some corrections of DOIs and formatting errors, which have also been solved. In the track-changed version, in References 2 and 38, their respective DOIs have been formatted. In the clean version and the track changes version, we also addressed it accordingly.

Harmonization was done between journal titles and citation style to comply with PLOS ONE guidelines.

These changes have been implemented in the revised manuscript and are visible in the tracked-changes file.

Additional Editor Comments

Comment:

The sample-size calculation assumes a 30% prevalence of vaping among university students in Bangladesh, described as anecdotal in the absence of available data. Could you clarify the scientific basis for selecting this value? Please indicate whether it was derived from published studies, pilot data, or a conservative estimation approach, and consider providing an appropriate reference or justification.

Response: Thank you for this important methodological question. We have clarified and strengthened the justification for the assumed 30% prevalence used in the sample size calculation. At the time of study design, no nationally representative data were available on vaping prevalence among Bangladeshi university students. Therefore, we adopted a conservative prevalence estimate informed by published evidence from comparable South Asian populations. We considered regional evidence from South Asia, where vaping prevalence among young adults and university students has been reported to be high. For example, a study from India reported a vaping prevalence of 23% among young people (please see lines 187-188 and page no. 7 on the track changes version). We have now revised the Methods (Section 2.3: Sampling) to explicitly state that “A conservative estimate of 30% prevalence of vaping was adopted based on published South Asian studies [8].” We have cited the relevant literature accordingly. Please see below.

Ref: 8: Pettigrew S, Santos JA, Miller M, Raj TS, Jun M, Morelli G, et al. E-cigarettes: A continuing public health challenge in India despite comprehensive bans. Prev Med Rep. 2023; 31:102108.

doi:10.1016/j.pmedr.2022.102108

---

## [Editor Report · Decision Letter 3]

8 Feb 2026

Vaping and mental health: A cross-sectional study among university students in Bangladesh

PONE-D-25-24214R3

Dear Dr. Farah Sabrina

We’re pleased to inform you that your manuscript has been judged scientifically suitable for publication and will be formally accepted for publication once it meets all outstanding technical requirements.

Kind regards,

Ali Awadallah Saeed

Academic Editor

PLOS One
---

## [Editor Report · Acceptance letter]

PONE-D-25-24214R3

PLOS One

Dear Dr. Sabrina,

I'm pleased to inform you that your manuscript has been deemed suitable for publication in PLOS One. Congratulations! Your manuscript is now being handed over to our production team.

Kind regards,

on behalf of

Dr. Ali Awadallah Saeed

Academic Editor

PLOS One